

# Comparison of hand grip strength and upper limb pressure pain threshold between older adults with or without non-specific shoulder pain

Cesar Calvo Lobo[1], Carlos Romero Morales[2], David Rodríguez Sanz[2], Irene Sanz Corbalán[3], Eleuterio A. Sánchez Romero[4], Josué Fernández Carnero[4,5,6] and Daniel López López[7]

[1] Physiotherapy Department, Motion in Brains Research Group, Instituto de Neurociencias y Ciencias del Movimiento, Centro Superior de Estudios Universitarios La Salle, Universidad Autónoma de Madrid, Madrid, Spain

[2] Physical Therapy & Health Sciences Research Group, Physiotherapy Department, Faculty of Health, Exercise and Sport, European University, Madrid, Spain

[3] School of Nursing, Physiotherapy and Podiatry, Universidad Complutense de Madrid, Madrid, Spain

[4] Department of Physical Therapy, Occupational Therapy, Rehabilitation and Physical Medicine, Rey Juan Carlos University, Madrid, Spain

[5] Research Group on Movement and Behavioural Science and Study of Pain, The Center for Advanced Studies University La Salle, Autónoma University, Madrid, Spain

[6] La Paz Hospital Institute for Health Research, IdiPAZ, Madrid, Spain

[7] Research, Health and Podiatry Unit, Department of Health Sciences, Faculty of Nursing and Podiatry, Universidade da Coruña, Ferrol, A Coruña, Spain

Corresponding authors
Cesar Calvo Lobo,
cecalvo19@hotmail.com
Carlos Romero Morales,
carlosmorales92@hotmail.com
David Rodríguez Sanz,
davidrodriguezsanz@gmail.com
Irene Sanz Corbalán, iresanzcorbalan@gmail.com
Eleuterio A. Sánchez Romero, elusanchezromero@gmail.com
Josué Fernández Carnero, josuefernandezcarnero@gmail.com
Daniel López López,
daniellopez@udc.es

## ABSTRACT

**Background.** There is a high prevalence of non-specific shoulder pain associated with upper limb functional limitations in older adults. The purpose of this study was to determine the minimal clinically important differences (MCID) of grip strength and pressure pain threshold (PPT) in the upper limb between older adults with or without non-specific shoulder pain.

**Methods.** A case-control study was carried out following the Strengthening the Reporting of Observational Studies in Epidemiology (STROBE) criteria. A sample of 132 shoulders (mean $\pm$ SD years) with ($n = 66$; $76.04 \pm 7.58$) and without ($n = 66$; $75.05 \pm 6.26$) non-specific pain were recruited. The grip strength and PPT of the anterior deltoid and extensor carpi radialis brevis (ECRB) muscles were assessed.

**Results.** There were statistically significant differences (mean $\pm$ SD; $P$-value) for anterior deltoid PPT ($2.51 \pm 0.69$ vs $3.68 \pm 0.65$, kg/cm$^2$; $P < .001$), ECRB PPT ($2.20 \pm 0.60$ vs $3.35 \pm 0.38$ kg/cm$^2$; $P < .001$) and grip strength ($20.78 \pm 10.94$ vs $24.63 \pm 9.38$ kg; $P = .032$) between shoulders with and without non-specific pain, respectively.

**Discussion.** The MCID of 1.17 kg/cm$^2$, 1.15 kg/cm$^2$ and 3.84 kg were proposed for anterior deltoid PPT, ECRB PPT and grip strength, respectively, to assess the upper limb of older adults with non-specific shoulder pain after treatment. In addition, univariate and multivariate (linear regression and regression trees) analyses may be used to consider age distribution, sex, pain intensity, grip strength and PPT in older adults including clinical and epidemiological studies with non-specific shoulder pain.

## INTRODUCTION

The aging population has a high prevalence of non-specific shoulder pain (31% with severe pain intensity) and associated functional limitations (36% with greater difficulty performing daily tasks associated with reduced internal rotation). This can have a considerable impact on public health. In addition, 41% of these subjects present with bilateral shoulder pain. This condition may be frequently overlooked and consequently undertreated (*Burner et al., 2014*).

Radiological findings are frequently not be correlated with shoulder pain in older adults. The cost of imaging shoulders in the elderly–especially magnetic resonance imaging–is significant, and the relevance of the findings is questionable (*Gill et al., 2014*). New treatments strategies for older adults with symptomatic shoulders are necessary (*Burner et al., 2014*). Therefore, non-specific shoulder pain is associated with myofascial pain syndrome (MPS), and treatment can reduce pain intensity, increase pressure pain threshold (PPT) and modify grip strength in the upper limbs of older adults (*Calvo, Pacheco & Hita, 2015*; *Calvo-Lobo et al., 2016*).

Furthermore, the MPS may be defined as a set of sensitive, motor or autonomic signs and symptoms originated by hyperirritable spots in a muscle taut band—these are myofascial trigger points (MTrPs) (*Dommerholt et al., 2016*). Active MTrPs produce spontaneous and recognized pain while latent MTrPs generate local or referred pain after stimulation (*Ge & Arendt-Nielsen, 2011*; *Gerber et al., 2013*). Indeed, the presence of latent MTrPs in the upper limb may influence the peripheral and central sensitization process and alter the strength and PPT in the upper limb of patients with shoulder pain (*Calvo, Pacheco & Hita, 2015*; *Calvo-Lobo et al., 2016*; *Ge et al., 2008*; *Celik & Yeldan, 2011*; *Coronado et al., 2014*). Nevertheless, the relationship between shoulder pain and PPT in the upper limb and grip strength should be more widely studied during aging.

On one hand, the aging process generates a non-statistically significant reduction in the PPT of the upper extremities across different age ranges (*Donat et al., 2005*). Indeed, *Neziri et al. (2011)* showed that age, sex, and/or the interaction of age with sex may affect the pressure pain measures. Furthermore, women are more sensitive to PPT than men. Despite this, the influence of sex reduces with age.

*Lautenbacher et al. (2005)* showed that PPT temporal summation may not be influenced by age. The low PPT perception in older adults shows that deep tissue (muscle) nociception may be less influenced by age than superficial tissue (skin) nociception. In conclusion, reference values need to be determined by body region, gender and age increases (*Neziri et al., 2011*).

On the other hand, an age-related grip strength reduction is shown in the elderly, although this reduction does not reach a statistical significant difference by aged distribution in this population (*Donat et al., 2005*). Moreover, *Abizanda et al. (2012)* reported that the grip strength values were almost double in men than in women older adults. *Bohannon (2008)* and *Norman et al. (2011)* showed that low grip strength may be associated with increases in premature mortality, disability, risk of complications, length of stay after

hospitalization or surgery, and nutritional alterations in older adults. Daily pain is common and is associated reduced grip strength in the elderly (*Landi et al., 2009*).

Grip strength and PPT in the satellite latent MTrPs of the anterior deltoid and extensor carpi radialis brevis (ECRB) muscles were assessed in the treatment of key active MTrPs in the infraspinatus muscle in older adults with non-specific shoulder pain. The treatment of the latent MTrP associated with the key active MTrP of the infraspinatus muscle improves pain intensity and increases the PPT of the satellite latent MTrPs located in the referred pain area over the short term (*Calvo, Pacheco & Hita, 2015*; *Calvo-Lobo et al., 2016*).

To date, the minimum clinical important differences (MCID) of the grip strength and PPT in the upper limb have not been established in this population. The aim of this study was to evaluate the MCID of the grip strength and PPT in the upper limbs between older adults with or without non-specific shoulder pain.

## MATERIAL & METHODS

### Design
A case-control study was carried out from October 2013 to March 2016 following the Strengthening the Reporting of Observational Studies in Epidemiology (STROBE) statement and checklist (*Von Elm et al., 2007*). Previously, the study was approved by the clinical research ethics committee of the general hospital from the health area. All subjects signed an informed consent form before their inclusion in the study. Furthermore, the Helsinki Declaration and ethical standards in human experimentation were followed.

### Participants
A sample of 132 shoulders (mean $\pm$ SD years) with ($n = 66$; $76.04 \pm 7.58$) and without ($n = 66$; $75.05 \pm 6.26$ years) non-specific pain were recruited from a care center and their homes.

The inclusion criteria were unilateral (for case group) or bilateral (for control group) shoulders from subjects aged 65 and over with or without non-specific shoulder pain. At least one latent MTrP in the anterior deltoid and one latent MTrP in the ECRB were required for each upper limb to justify their PPT and grip strength assessments. For the case group, both latent MTrPs were located ipsilateral to the painful shoulder. For the control group, these latent MTrPs were situated bilateral in both non-painful shoulders. Shoulder pain was considered if there was pain at rest and mainly located in the glenohumeral joint between the acromion, the insertion of the deltoid muscle, and the lateral region of the scapula (*Calvo, Pacheco & Hita, 2015*; *Calvo-Lobo et al., 2016*). Non-specific shoulder pain was considered if a previous diagnosis was not present in the medical record considering structural, neurological, visceral or red flag conditions (*Mitchell et al., 2005*). A latent MTrP was diagnosed if there were palpable tender nodules in a taut band without the patient's local or referred pain recognition. If there was more than one latent MTrP in the same muscle, then the MTrP most hyperalgesic to palpation was selected as the one that generated the highest pain intensity in the Numeric Rating Scale (NRS) under the same pressure (*Calvo, Pacheco & Hita, 2015*; *Calvo-Lobo et al., 2016*). The convention for PPT assessment in musculoskeletal conditions is the use of a designated site rather than the

most painful MTrP (*Coronado et al., 2014*). However, we prioritized the most hyperalgesic latent MTrPs due to their relationship with the PPT reduction secondary to peripheral and central sensitization as well as strength modifications in the upper limb (*Calvo, Pacheco & Hita, 2015*; *Calvo-Lobo et al., 2016*; *Ge et al., 2008*; *Celik & Yeldan, 2011*).

The exclusion criteria were prior diagnoses or treatments in the medical record for myopathy, neuropathy, cognitive impairments, joint conditions (cervical, rotator-cuff or glenohumeral regions), surgeries (upper-limb or cervical regions), physiotherapy treatment (within the previous six months), corticoid injections (within the previous one year), analgesic or anti-inflammatory medication (within the previous one week) (*Calvo, Pacheco & Hita, 2015*; *Calvo-Lobo et al., 2016*).

## Outcome measures

A physical therapist with the necessary specialization and experience and good interexaminer reproducibility ($\kappa = 0.63$) in the MTrPs clinical evaluation according to *Myburgh et al. (2011)*, carried out all of the assessments. Socio-demographic (age and sex), pain intensity, anterior deltoid, ECRB PPT, and grip strength were measured. Pain intensity ipsilateral to the painful shoulder was only assessed in the case group using the NRS, which is recommended for a reliable and valid use in older adults (*Taylor et al., 2005*).

### *Primary outcome*

PPT was assessed from 0 to 10 kg/cm$^2$ with a mechanical algometer (FDK/FDN, Wagner Instruments, Greenwich, CT). This is reliable, reproducible and sensitive for latent MTrPs evaluation in older adults' upper limbs (*Donat et al., 2005*; *Neziri et al., 2011*; *Lautenbacher et al., 2005*; *Koo, Guo & Brown, 2013*; *Fisher, 1998*; *Park et al., 2011*). The most hyperalgesic latent MTrP in the anterior deltoid and the most hyperalgesic latent MTrP in the ECRB ipsilateral to the painful shoulder (for the case group) or bilateral for the non-painful shoulders (for the control group) were assessed. The procedure (mean of three repeated measurements with a 30–60 s interval) and position (supine decubitus with the forearm on the abdomen) were followed according to prior studies (*Calvo, Pacheco & Hita, 2015*; *Calvo-Lobo et al., 2016*).

### *Secondary outcome*

The maximum grip strength was performed from 0 to 90 kg with a hydraulic hand dynamometer (JAMAR, Sammons Preston Rolyan, Sammons Court Bolingbrook, IL). This is a valid, reproducible and reliable tool in older adults (*Abizanda et al., 2012*; *Bohannon, 2008*; *Norman et al., 2011*). Coinciding with the PPT assessment, the painful side for the case group and both upper limbs for the control group were assessed by grip strength. A single assessment (5–10 s) used the elbow flexed at 90° with neutral pronation or supination as well as a sitting position (*Calvo, Pacheco & Hita, 2015*; *Calvo-Lobo et al., 2016*; *Abizanda et al., 2012*).

## Intraexaminer reliability

An additional intraexaminer reliability study for the latent MTrP localization procedure coinciding with skin marks was carried out in the upper limb of 42 older adults. The most

hyperalgesic latent MTrP was selected at 2 different moments with an interval of at least 1 h using the same position described for the PPT assessment. Upon the first assessment, the skin was previously marked at the location of the most hyperalgesic latent MTrP in each muscle. The second assessment used the prior marks that coincided with the displaced skin with each one of the most hyperalgesic latent MTrPs. For the anterior deltoid, the distance (cm) from the most hyperalgesic latent MTrP to the lateral angle of the acromion was performed. For the ECRB, the distance from the most hyperalgesic latent MTrP to the lateral epicondyle of the elbow was measured. The reason that the previous skin marks were used was because they facilitate a procedure with better intraexaminer reliability for future evaluations during treatments (*Calvo, Pacheco & Hita, 2015*; *Calvo-Lobo et al., 2016*).

## Data analysis

SPSS version 22.0 (IBM SPSS Statistics for Windows; Armonk, NY: IBM Corp) and $\alpha$ error of 0.05 (95% confidence interval) and a desired power of 80% ($\beta$ error of 0.2) was used for statistical analysis. The Shapiro-Wilk test and the visual distribution did not show large normality deviations. Parametric analysis was used because of this and the large sample size.

Afterwards, a comparison of both the sociodemographic data and the main outcomes was performed between case and control groups. For case and control groups and sex, comparisons used Fisher's exact test. Age was divided into three intervals to permit a similar number of shoulders in each group (from 65 to 70, from 71 to 80, and more than 80 [81–97.2] years); Pearson's Chi-square test compared between the case and control groups. Furthermore, Student's $t$-test for independent samples was used for age and outcomes (anterior deltoid and ECRB PPT, and grip strength) and the sex and age group. The minimum clinical differences of the anterior deltoid and ECRB PPT and grip strength were calculated by subtracting the case group from the control group means according to prior studies (*Lobo et al., 2016a*; *Lobo et al., 2016b*). Box plots were used to illustrate PPT and grip strength values from the case and control groups.

A univariate correlation analysis using Pearson's ($r$) coefficient was performed to evaluate the relationship between the anterior deltoid and ECRB PPT as well as the grip strength measurements of the groups. Correlations were interpreted as weak (0.00–0.40), moderate (0.41–0.69), or strong (0.70–1.00) (*Hoch et al., 2015*).

In addition, a multivariate predictive analysis used linear regression and regression trees. Linear regression was performed via a stepwise selection method and the $R^2$ coefficient to establish quality adjustments. Sociodemographic data including age, sex (male = 0; female = 1), and group (control = 0; case = 1) as well as the pain intensity were used as independent variables. The anterior deltoid and ECRB PPT and grip strength were dependent variables.

Finally, the intraexaminer reliability for the latent MTrP localization procedure was added according to *Bland & Altman (2010)*. First, the normality, descriptive analyses and repeated-measures ANOVA were calculated. The intraclass correlation coefficient (ICC) used a two-factor model with mixed effects (ICC A) with absolute agreement and a one-factor model with random effects (ICC B). Cronbach's alpha and the standard error of measurement (SEM) were calculated. In addition, the minimum detectable

**Table 1 Sociodemographic and pain intensity characteristics of the sample.**

| Sociodemographic data and pain intensity characteristics | | Total sample (n = 132) | Painful shoulders (n = 66) | Non-painful shoulders (n = 66) | P-value case vs. control |
|---|---|---|---|---|---|
| Age mean (SD) | ≥65 years | 75.54 (6.86) | 76.04 (7.58) | 75.05 (6.26) | .415[a] |
| Age distribution n(%) | 65–70 years | 34 (25.8) | 18 (27.3) | 16 (24.2) | |
| | 71–80 years | 60 (45.5) | 28 (42.4) | 32 (48.5) | .783[b] |
| | >80 years | 38 (28.8) | 20 (30.3) | 18 (27.3) | |
| Sex n(%) | Men | 51 (38.6) | 23 (34.8) | 28 (42.4) | .475[c] |
| | Women | 81 (61.4%) | 43 (65.2) | 38 (57.6) | |
| Pain intensity mean(SD) | 65–70 years | N/A | 4.94 (1.80) | N/A | |
| | 71–80 years | N/A | 4.93 (1.58) | N/A | N/A |
| | >80 years | N/A | 4.80 (1.85) | N/A | |
| | Men | N/A | 4.48 (1.62) | N/A | N/A |
| | Women | N/A | 5.12 (1.72) | N/A | |

**Notes.**

Abbreviations:: N/A, Not applicable.

[a]Student's *t* test for independent samples.
[b]Pearson chi-square test.
[c]Fisher's exact test.

change (MDC) was calculated for 95%, 90% and 80% confidence intervals (CI) using the $\sqrt{2}$×1.96×*sw*; $\sqrt{2}$×1.645×*sw* and $\sqrt{2}$×1.28×*sw* equations, respectively. The heteroscedasticity was performed by the Kendall *tau_b* ($\tau$B) coefficient to validate this method. Bland and Altman graphs were prepared including the agreement limits and the linear regression analysis between the means and the differences (dependent variable) of the measurements (*Bland & Altman, 2010*; *Stratford & Goldsmith, 1977*).

## RESULTS

The sex and age distribution were homogeneous between case and control groups (Table 1). There were statistically significant differences (mean $\pm$ SD; *P*-value) for anterior deltoid PPT ($2.51 \pm 0.69$ vs $3.68 \pm 0.65$, kg/cm$^2$; $P < .001$), ECRB PPT ($2.20 \pm 0.60$ vs $3.35 \pm 0.38$ kg/cm$^2$; $P < .001$) and grip strength ($20.78 \pm 10.94$ vs $24.63 \pm 9.38$ kg; $P = .032$) between shoulders with and without non-specific pain, respectively. The MCID of 1.17 kg/cm$^2$, 1.15 kg/cm$^2$ and 3.84 kg were proposed for anterior deltoid PPT, ECRB PPT and grip strength, respectively, to treat older adults with non-specific shoulder pain (Figs. 1–3).

According to the statistically significant differences ($P < .01$) of the univariate correlation analysis (Table 2), there was a positive correlation from strong ($r = 0.806$) to moderate ($r = 0.590$ to $0.677$) between the anterior deltoid and ECRB PPT. For grip strength of the groups, there was a moderate or weak correlation with the age ($r = -0.327$ to $-0.488$), anterior deltoid ($r = 0.315$ to $0.372$) and ECRB ($r = 0.303$ to $0.442$) PPT. The pain intensity did not present any statistically significant difference ($P > .05$).

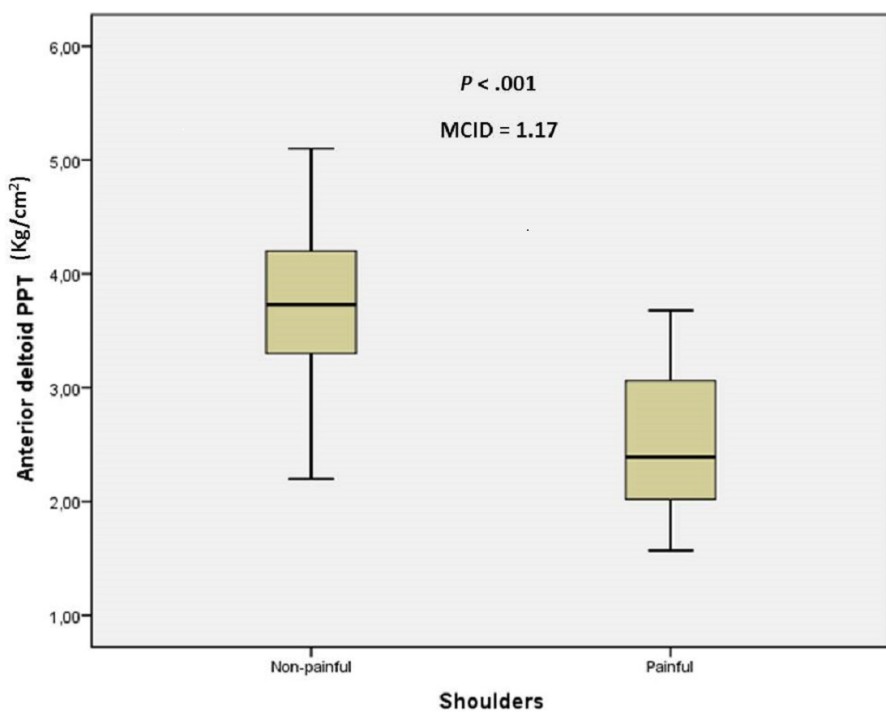

**Figure 1    Box plots to illustrate anterior deltoid PPT values between shoulders with and without pain.**
Abbreviations: MCID, minimal clinically important differences; PPT, pressure pain threshold.

**Table 2    Univariate correlation of the case-control groups and total sample.**

| Pearson's ($r$) coefficient[a] | | Total sample ($n = 132$) | Painful shoulders ($n = 66$) | Non-painful shoulders ($n = 66$) |
|---|---|---|---|---|
| Age | Anterior deltoid PPT | −0.044 | −0.021 | 0.037 |
| | ECRB PPT | −0.009 | 0.015 | 0.170 |
| | Grip strength | −0.327[*] | −0.488[*] | −0.083 |
| | Pain intensity | N/A | −0.003 | N/A |
| Anterior deltoid PPT | ECRB PPT | 0.806[*] | 0.590[*] | 0.677[*] |
| | Grip strength | 0.315[*] | 0.161 | 0.372[**] |
| | Pain intensity | N/A | −0.144 | N/A |
| ECRB PPT | Grip strength | 0.303[*] | 0.151 | 0.442[*] |
| | Pain intensity | N/A | −0.146 | N/A |
| Grip strength | Pain intensity | N/A | −0.109 | N/A |

**Notes.**
   Abbreviations:: ECRB, extensor carpi radialis brevis; PPT, pressure pain threshold; N/A, Not applicable.
   [a]Correlations were interpreted as weak (0.00–0.40), moderate (0.41–0.69) or strong (0.70–1.00) (*Mitchell et al., 2005*).
   [*]$P < 0.001$.
   [**]$P = 0.002$.

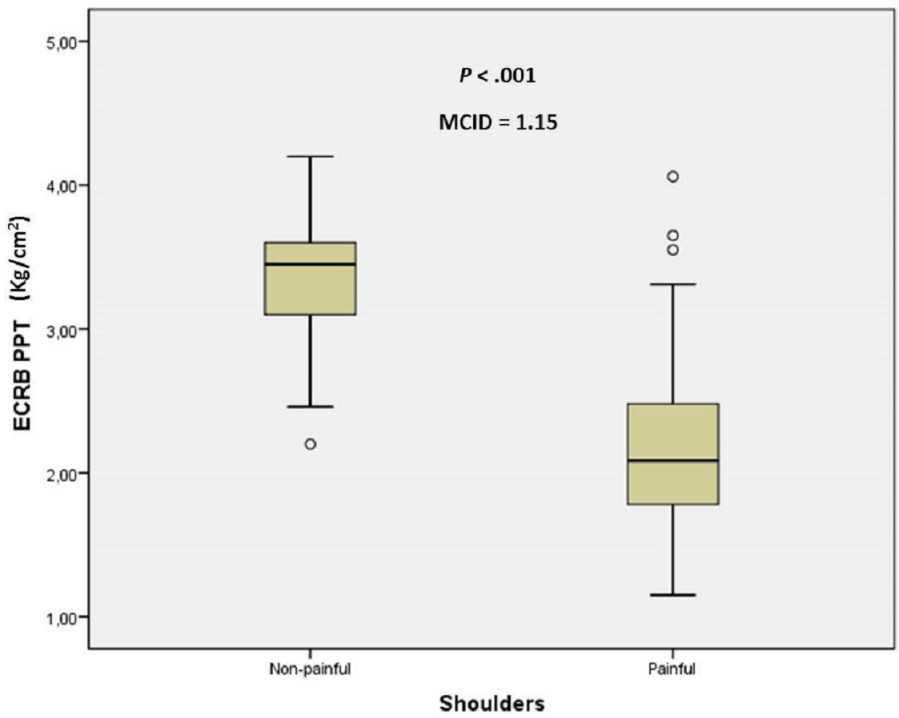

**Figure 2** **Box plots to illustrate ECRB PPT values between shoulders with and without pain.** Abbreviations: ECRB, extensor carpi radialis brevis; MCID, minimal clinically important differences; PPT, pressure pain threshold.

The linear regression model (Table 3) showed significant differences ($P < .05$), and the model $R^2$ varied from 0.012 to 0.565. In addition, the regression trees (Fig. 4) established 3–4 statistically significant nodes ($P < .05$).

Finally, the MTrP localization procedure showed high intraexaminer reliability after one hour of assessment (Table 4). In the anterior deltoid muscle, the most hyperalgesic latent MTrP was localized $7.07 \pm 1.13$ (4.55–9.85) cm to the lateral angle of the acromion. In the ECRB muscle, the most hyperalgesic latent MTrP was $6.06 \pm 0.82$ (4.75–7.88) cm from the lateral epicondyle. A satisfactory randomness within the limits of agreement was seen in the Bland-Altman plots (Fig. 5).

## DISCUSSION

The proposed MCID for the grip strength, anterior deltoid and ECRB PPT established the upper limb relevant clinical change in older adults with non-specific shoulder pain. Despite the presence of statistically significant differences, some interventions did not reach the MCID for PPT and grip strength in this condition and population (*Calvo, Pacheco & Hita, 2015*; *Calvo-Lobo et al., 2016*).

Clinically, the findings of this study support the usefulness of the upper limb PPT and grip strength assessments in older adults with non-specific shoulder pain. Nevertheless, some aspects should be considered in these evaluations. First, the latent MTrP PPT of the ECRB presented a larger difference percentage (34.32%) than the anterior deltoid

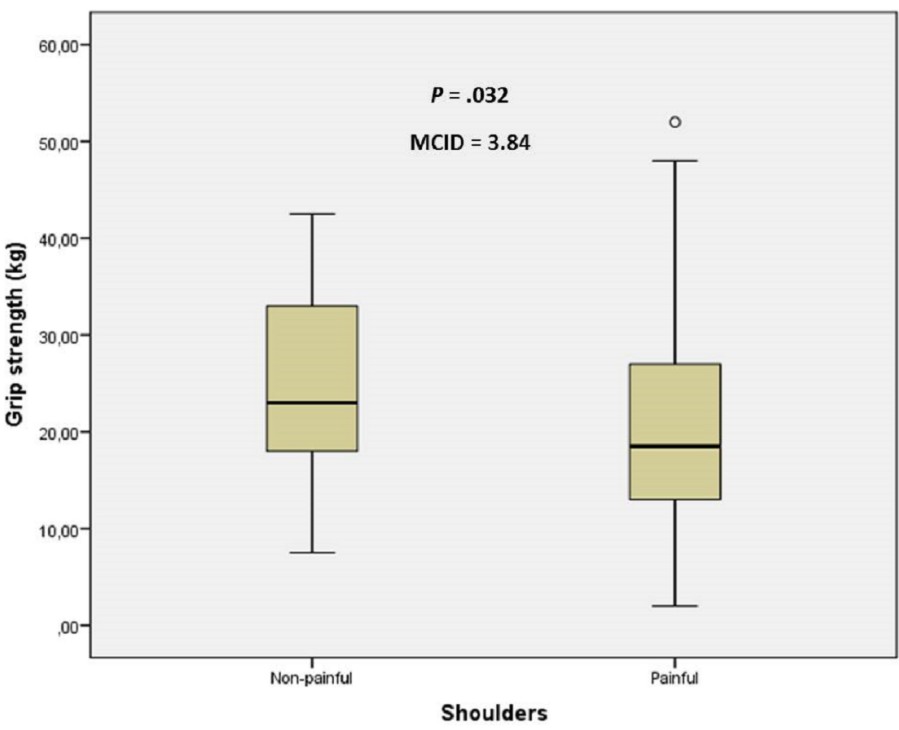

**Figure 3** **Box plots to illustrate grip strength values between shoulders with and without pain.** Abbreviations: MCID, minimal clinically important differences.

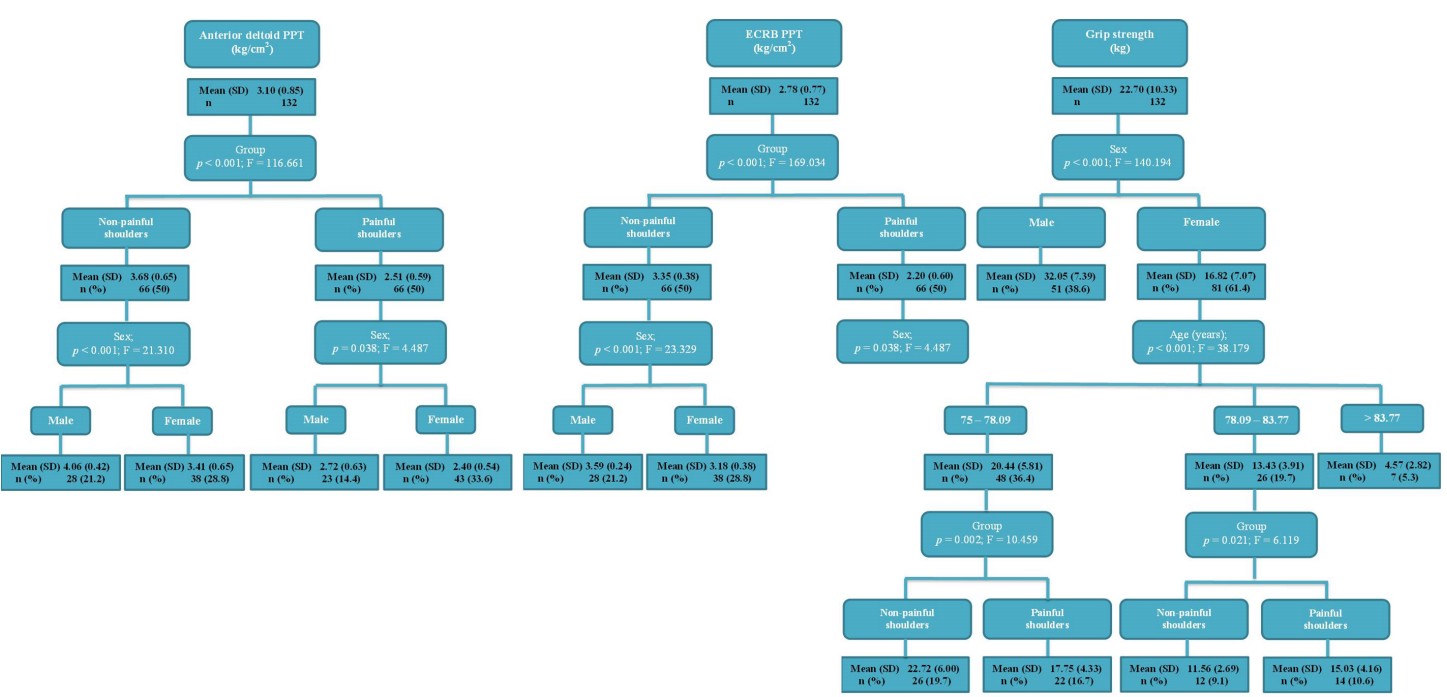

**Figure 4** **Regression tree nodes to predict the grip strength, anterior deltoid and ECRB PPT.** Abbreviations: ECRB, extensor carpi radialis brevis; PPT, pressure pain threshold.

**Table 3  Multivariate predictive analysis of the grip strength, anterior deltoid and ECRB PPT.**

| Parameter | Model | $R^2$ change | Model $R^2$ |
|---|---|---|---|
| PPT (kg/cm$^2$) | | | |
|     Anterior deltoid | 3.973 | | |
| | $-1.138^a$ group | $0.473^*$ | |
| | $-0.493^a$ sex | $0.078^*$ | 0.551 |
|     ECRB | 3.559 | | |
| | $-1.129^a$ group | $0.565^*$ | |
| | $-0.348^a$ sex | $0.048^*$ | 0.614 |
| Grip Strength (kg) | | | |
| | 69.079 | | |
| | $-15.070^a$ sex | $0.519^*$ | |
| | $-0.477^a$ age | $0.108^*$ | |
| | $-2.228^a$ group | $0.012^{**}$ | 0.638 |

Notes.

    Abbreviations:: ECRB, extensor carpi radialis brevis; PPT, pressure pain threshold.

    [a] Multiplay: age (years); sex (male = 0; female = 1); group (control = 0; case = 1).

    *$P$ value < 0.001.

    **$P$ value = 0.045.

**Table 4  Intraexaminer reliability for the latent MTrP localization.**

| Intraexaminer reliability | Anterior deltoid latent MTrP | ECRB latent MTrP |
|---|---|---|
| Repeated measures ANOVA $P$-value | .114 | .646 |
| Cronbach's alpha | 0.994 | 0.990 |
| ICC A[a] (CI 95%) $P$-value | 0.98 (0.978–0.994) $P < .001$ | 0.98 (0.963–0.989) $P < .001$ |
| ICC B[b] (CI 95%) $P$-value | 0.98 (0.978–0.994) $P < .001$ | 0.98 (0.963–0.989) $P < .001$ |
| SEM (cm) | 0.122 | 0.118 |
| MDC[c] (cm) (CI 95, 90, 80%) | 0.338/0.284/0.221 | 0.327/0.275/0.214 |
| Kendall $tau\_b$ ($\tau$B) $P$-value | $-0.101$ $P = .360$ | $-0.037$ $P = .735$ |
| $R^2$ $P$-value | 0.056 $P = .131$ | 0.005 $P = .669$ |

Notes.

    Abbreviations:: ANOVA, Analysis of variance; CI, Confidence interval; ECRB, extensor carpi radialis brevis; ICC, Intraclass correlation coefficient; MDC, Minimum detectable change; MTrP, Myofascial trigger point; PPT, pressure pain threshold; SEM, Standard error of measurement.

    [a] ICC A was calculated by means of a two-factor model with mixed effects.

    [b] ICC B was performed by means of one-factor model with random effects.

    [c] MDC was determined for each CI: $\sqrt{2} \times 1.96 \times sw$ (95% CI); $\sqrt{2} \times 1.645 \times sw$ (90% CI); $\sqrt{\sqrt{2}} \times 1.28 \times sw$ (80% CI).

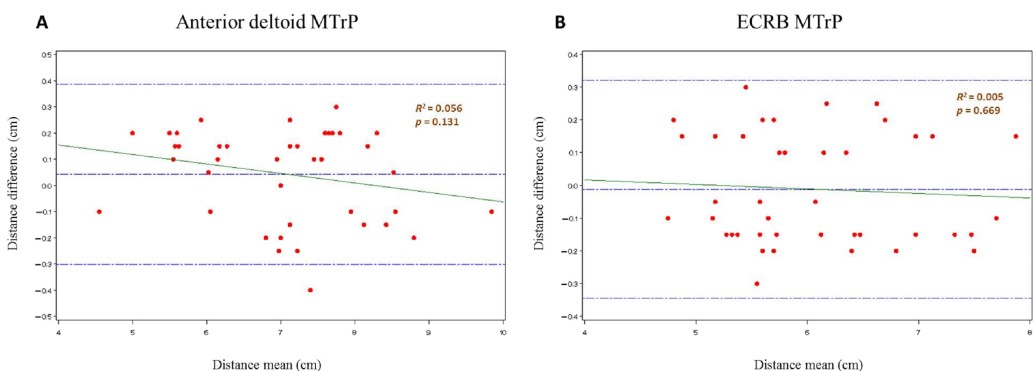

**Figure 5  Bland and Altman[2] graphs, completed with linear regression analysis, of the intra-examiner reliability of the anterior deltoid (A) and ECRB (B) latent MTrPs localization procedure.** Abbreviations: ECRB, extensor carpi radialis brevis; MTrPs, myofascial trigger points.

(31.79%) in painful shoulders with respect to non-painful shoulders. Second, the PPT and grip strength are influenced by the presence of shoulder pain and sex. Nevertheless, the grip strength may only be modified by the age distribution of older adults. Despite the existence of moderate $kg/cm^2$ intensity (*Taylor et al., 2005*), neither PPT nor grip strength were influenced by the pain intensity of the shoulders. Finally, this is the first study to show a positive correlation between the PPT and grip strength in non-painful shoulders although there is no statistically significant correlation in the painful shoulders. Further research is needed to investigate this field.

The PPT minimal detectable increase was proposed as 0.54 $kg/cm^2$ according to *Koo, Guo & Brown (2013)*. While *Fisher (1998)* determined a critical PPT MCID of 2 $kg/cm^2$ between a normal control point and MTrPs in the general population, *Fernández-Carnero et al. (2007)* showed that 2.4 $kg/cm^2$ was the MCID in subjects with lateral epicondylalgia. The MCID of this study varied from 1.15 to 1.17 $kg/cm^2$. This lower MCID may be because this research was performed between latent MTrPs of subjects with and without non-specific shoulder pain. In addition, older adults have low PPT perception (*Neziri et al., 2011*). The age-related change in pain-evoked activity may produce a functional reduction of brain areas (contralateral putamen and caudate) related to pain modulatory mechanisms with older age (*Cole et al., 2010*). Therefore, the MCID should be established for each type of condition, body region, sex and age (*Donat et al., 2005*; *Neziri et al., 2011*; *Lautenbacher et al., 2005*).

The grip strength MCID was 3.84 kg in older adults with non-specific shoulder pain. Nevertheless, the MCID proposed by *Dhara et al. (2009)* was higher and varied from 6.08 to 19.08 kg in patients with orthopedic conditions (stroke, arthritis and accidents) in the upper limb. This difference may be secondary to the condition type and the age range from 20 to 40 years. In older adults, the grip strength coincided with the reference values proposed by *Abizanda et al. (2012)*, *Bohannon (2008)* and *Cuesta-Vargas & Hilgenkamp (2015)*. In addition, the standard approach to grip strength measurements was followed *Abizanda et al. (2012)* and *Roberts et al. (2011)*.

In addition, univariate (Table 2) and multivariate (Table 3 and Fig. 4) analyses may be considered with respect to age distribution, sex, pain intensity, grip strength and PPT in older adults clinical and epidemiological research with non-specific shoulder pain. The linear regression and regression trees show similar conclusions. Furthermore, the regression trees add some important considerations. First, only gender influences the ECRB PPT in the control group. Second, the age distribution only affects the grip strength assessment women.

A high intraexaminer reliability was seen because the MTrPs localization procedure was carried out coinciding with the most hyperalgesic latent MTrP with prior skin marks to justify the PPT assessment point at two different times (*Calvo, Pacheco & Hita, 2015*; *Calvo-Lobo et al., 2016*). Despite this, the $R^2$ coefficient, SEM and MDC (CI 95%) was lower in the ECRB MTrP (0.005, 0.118 cm and 0.327 cm) than the anterior deltoid MTrP (0.056, 0.122 cm and 0.338 cm) location (Table 4 and Fig. 5), respectively. Nevertheless, the interexaminer reliability was not performed because all assessments were performed by the same rater with the MPS experience necessary to achieve a good interexaminer agreement in the MTrPs location in relation to shoulder pain according to *Myburgh et al. (2011)*.

Several limitations should be considered. Despite the wide variability of methods used to calculate the MCID (*Chung et al., 2016*), we used a case-control study (*Lobo et al., 2016a*; *Lobo et al., 2016b*). Age intervals were not equal by years' distribution in order to permit a similar number of shoulders in each group (Table 1). In healthy older adults, the neuronal circuits of cognitive inhibition and conscious pain control may overlap and interfere with the PPT assessment (*Zhou et al., 2015*). Sympathetic responses are associated with local and referred hyperalgesia of the MTrPs in subjects with shoulder pain, and this may alter the PPT evaluation (*Ge, Fernández-de-las-Peñas & Arendt-Nielsen, 2006*). Moreover, the grip strength reference value t-scores were not considered for determining dynapenia in the elderly (*Bohannon & Magasi, 2015*). The grip strength evaluation was not divided into dominant and non-dominant upper limbs to establish the MCID for both sides. This assessment might be disturbed by various comorbid conditions in older adults (*Abizanda et al., 2012*; *Bohannon, 2008*; *Norman et al., 2011*; *Cuesta-Vargas & Hilgenkamp, 2015*; *Roberts et al., 2011*; *Bohannon & Magasi, 2015*).

## CONCLUSIONS

The MCID increase of the PPT and grip strength is proposed to assess the upper limb after interventions in older adults with non-specific shoulder pain. In addition, univariate and multivariate (linear regression and regression trees) analyses may be used to consider age distribution, sex, pain intensity, grip strength and PPT in older adult's clinical and epidemiological studies with non-specific shoulder pain.

### Funding

The authors received no funding for this work.

## Competing Interests

The authors declare there are no competing interests.

## Author Contributions

- Cesar Calvo Lobo conceived and designed the experiments, performed the experiments, analyzed the data, contributed reagents/materials/analysis tools, wrote the paper, prepared figures and/or tables, reviewed drafts of the paper.
- Carlos Romero Morales, David Rodríguez Sanz, Irene Sanz Corbalán and Eleuterio A. Sánchez Romero performed the experiments, wrote the paper, prepared figures and/or tables, reviewed drafts of the paper.
- Josué Fernández Carnero analyzed the data, wrote the paper, prepared figures and/or tables, reviewed drafts of the paper.
- Daniel López López analyzed the data, contributed reagents/materials/analysis tools, wrote the paper, prepared figures and/or tables, reviewed drafts of the paper.

## Human Ethics

The following information was supplied relating to ethical approvals (i.e., approving body and any reference numbers):

The study was approved by the Clinical Intervention Ethics Committee of the General Hospital from Segovia (10/2013). All participants gave written informed consent before data collection began.

## Data Availability

The raw data has been supplied as a Supplementary File.

## Supplemental Information

Supplemental information for this article can be found online at http://dx.doi.org/10.7717/peerj.2995#supplemental-information.

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
