# Peer review of "Comparison of hand grip strength and upper limb pressure pain threshold between older adults with or without non-specific shoulder pain"

_PeerJ, doi:10.7717/peerj.2995_

## Round 0.1 · original submission · Minor Revisions

We thank you for your submission of this manuscript at PeerJ. The three reviewers are positive about the strengths of the manuscript, but have also identified a number of more minor amendments which should be attended to prior to resubmission. Of particular note, all three reviewers have identified aspects of the written English as needing improvement. I would therefore recommend you get additional assistance on this aspect of the manuscript.

·

Basic reporting

• The English language could be improved in some sections to ensure an international audience can clearly understand your text. Some examples where language could be improved include lines 68, 82, 87, 119, 133, 144, 224, 225.
• The introduction and background do not build a strong case for assessing PPT, you need to better identify/outline links between PPT and pain/sensitization/dysfunction (from line 79) in musculoskeletal pain and specifically upper limb pain. The case for grip strength is stronger as you have linked it with mortality and disability (line 91).
• The discussion should identify the importance of the findings, differences between painful and non-shoulder patients and how these differences can effect clinical outcomes/assessment (if applicable).
• The discussion should also include information on the intra-examiner reliability
• Literature is well referenced but there should be more pain physiology/pain processing information within the introduction as it is introduced in the discussion but without prior context.

Experimental design

• The research question is appropriate but the case hasn’t been built in the introduction to justify the question/s relating to PPT
• Methods relating to assessment of PPT and grip strength need to be clarified, particularly for inclusion criteria (lines 118-120).
• Why is a MTrP used as the PPT assessment point when it is convention in musculoskeletal conditions to use a designated site rather than the most painful MTrP? Need to justify why MTrP used as assessment sites.
• Was there any other inclusion criteria for shoulder pain – having a latent MTrP is not a clear diagnosis of shoulder pain. Did they have shoulder pain at rest or with certain activities/movements?
• What elbow position is used for grip strength? Line 152
• Was grip strength assessed on the dominant arm?
• Be specific with age of >80 year group, are they aged 80-90? 80-85 years?
• Perhaps age ranges should be equal ie. 5 or 10 year age brackets
• Outline the importance/reason for the intra-examiner reliability and clarify the procedure regarding skin marks. Does this mean that the skin was marked before assessment or does this relate to anatomical land marks?

Validity of the findings

No comments

Additional comments

• The authors have investigated a growing area of research by assessing PPT and grip strength differences in those with and without shoulder pain. I commend the authors on their large data set of 132 shoulders which is a very high number for PPT in particular. The results regarding the differences in PPT and grip strength are of interest.
• Further clarification of the diagnosis of non-specific shoulder pain should be included to better support this important area of research
• The authors could make a stronger case for the need to have MDIC for PPT in non-specific shoulder pain by better identifying how PPT relates to pain/dysfunction in shoulder pain/upper limb pain

Reviewer 2 ·

Basic reporting

There are some grammatical issues with the paper, as follows:
Line 69: where did these percentages come from and what do they represent?
Line 73: is there any data regarding the cost burden?
Line 83: "Lautenbacher et al.7 researched that PPT temporal summation may not be
critically affected by age" - needs re-worded
Line 95: what were the findings of this?
Line 100 (and thereafter): should read 'established' not 'stablished'?
Line 104: Matherial - should be material?
Line 225: 'Despite of' - needs re-worded (also line 82)

Experimental design

Overall the experimental design was good. Research question is well designed and methods carried out with sufficient detail and information to replicate.
However, there does not appear to be any justification for the method used for calculation of minimal clinical differences.

Validity of the findings

No comment other than that already made regarding the calculation of minimal clinical differences.

Additional comments

Overall a sound study with some attention required regarding wording/grammar. I would like further justification, as previously outlined, for the minimal clinical differences calculation method.

Reviewer 3 ·

Basic reporting

There is need for basic editing of the entire document, as there are grammatical errors present.
e.g.
Line 44-45: no commas necessary &
Line 100, 182, 214, 224, & 236: written 'stablished' instead of established.

Experimental design

No comment.

Validity of the findings

No comment.

---

## Round 0.2 · accepted · Accept

We thank you for attending to the reviewers comments on your revised manuscript and am happy to let you know the paper has now been accepted for publication.

·

Basic reporting

The English has been improved and the authors have responded to reviewer feedback.

Experimental design

No comment

Validity of the findings

No comment

Additional comments

The revised manuscript is much improved and is easier to follow/read than the original manuscript. I commend the authors on their revisions.

Reviewer 2 ·

Basic reporting

Much improved English throughout. Sufficient references and background / context provided. All data presented in a professional manner.

Experimental design

Clear aims, well defined research question. Methods described with sufficient detail to replicate.

Validity of the findings

Statistically sound data, conclusion linked to research question. Limitations of study clearly identified.

Additional comments

This paper is much improved from the original submission.